# Risk Assessment of Pipeline Engineering Geological Disaster Based on GIS and WOE-GA-BP Models

**Bohu He [1], Mingzhou Bai [1,2,*], Hai Shi [1,2], Xin Li [1], Yanli Qi [1] and Yanjun Li [1]**

[1] School of Civil Engineering, Beijing Jiaotong University, Beijing 100044, China; 18115063@bjtu.edu.cn (B.H.); shihai@bjtu.edu.cn (H.S.); 16115268@bjtu.edu.cn (X.L.); 20115066@bjtu.edu.cn (Y.Q.); 21115070@bjtu.edu.cn (Y.L.)

[2] Key Laboratory of Track Engineering, Beijing Jiaotong University, Beijing 100044, China

* Correspondence: mzhbai@bjtu.edu.cn

**Featured Application: The consistency between the weight of each factor and the actual situation should be further studied.**

**Abstract:** Oil and gas pipelines are part of long-distance transportation projects which pass through areas with complex geological conditions and which are prone to geological disasters. Geological disasters significantly affect the safety of pipeline operations. Therefore, it is essential to conduct geological disaster risk assessments in areas along pipelines to ensure efficient pipeline operation, and to provide theoretical support for early warning and forecasting of geological disasters. In this study, the pipeline routes of the Sichuan-Chongqing and Western Hubei management offices of the Sichuan-East Gas Transmission Project were studied. Seven topographic factors—surface elevation, topographic slope, topographic aspect, plane curvature, stratum lithology, rainfall, and vegetation coverage index—were superimposed using the laying method with a total of eight evaluation indicators. The quantitative relationships between the factors and geological disasters were obtained using the geographic information system (GIS) and weight of evidence (WOE). The backpropagation neural network (BP) was optimised using a genetic algorithm (GA) to obtain the weight of each evaluation index. The quantified index was then utilized to identify the geological hazard risk zone along the pipeline. The results showed that the laying method, stratum lithology, and normalised difference vegetation index were the factors influencing hazards.

**Keywords:** oil and gas pipelines; hazard zone; GIS; GA; BP; WOE

## 1. Introduction

The oil and gas pipelines in China have undergone 17 years of large-scale construction; they were completed and the first west–east gas transmission pipeline was ready for operation in 2014. By the end of 2020, the total length of the pipelines reached 144,000 km, ranking fourth in the world (Figure 1). According to incomplete statistics, 2100 hidden danger points of geological hazards threatening pipeline safety have been discovered in the main oil and gas pipelines currently in service or under construction, of which 600 landslides accounted for nearly 30% of the hazards [1].

Unlike roads and railways, oil and gas pipelines transport flammable and explosive materials; therefore, if an accident occurs, the consequences are often more serious. For example, on 26 May 2013, the natural gas pipeline in the Yanggutang section of the Chishan Township of Xiangtan broke and ignited, causing minor injuries to two people. On 20 July 2016, the pipeline at the Enshi Yuanjiwan tunnel exit of the Sichuan natural gas pipeline to the east was broken by a landslide, resulting in a natural gas explosion; two people died and three were injured. In addition, an area of 39.65 m$^2$ was severely burned. On 2 July 2017, the highway slope of the China–Myanmar natural gas pipeline in Qinglong County, Guizhou province, collapsed and slid due to continuous rainfall which broke

the gas pipeline buried along the slope, resulting in gas leakage, a combustion, and an explosion. This event resulted in one death, 23 injuries, and the direct economic loss of 21.45 million yuan. To minimise the damage to oil and gas pipelines caused by landslides and other geological disasters, scientific and effective methods must be used to conduct risk assessments along pipelines, and to provide a theoretical basis for the detection and early warning of damage to oil and gas pipelines.

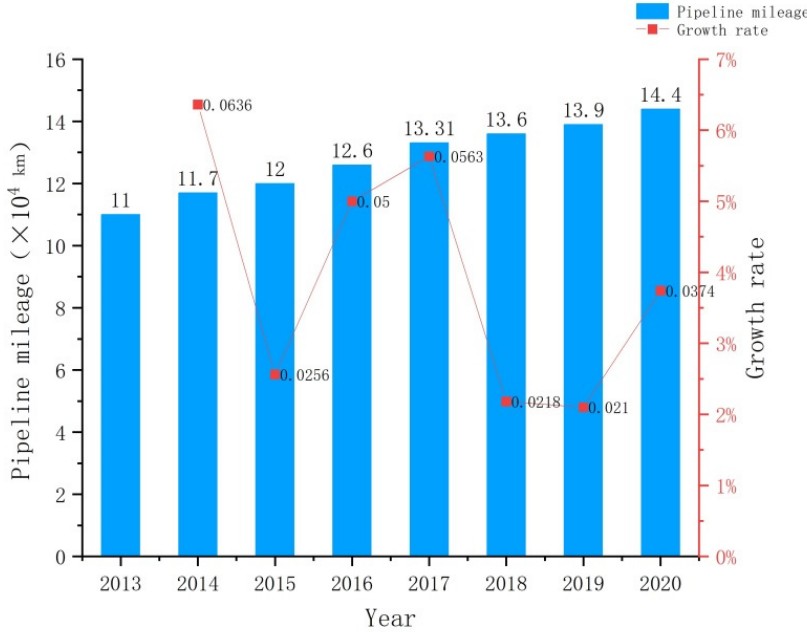

**Figure 1.** Pipeline mileage and growth rate in 2013–2021.

In recent years, many scholars have conducted research on pipeline stress and strain, limit state [2–5], monitoring, and early warning [6,7] during landslides; most of them focusing on pipeline stress analysis without considering the mechanical behaviour of different laying methods through quantitative analyses in pipeline risk assessments. Pipeline risk assessments include two aspects: a vulnerability assessment of geological hazards along the pipeline and a vulnerability assessment of the pipeline.

Risk assessments are affected by many factors; however, the dimensions, magnitudes, and evaluation indicators of each factor are different. To determine the accuracy of the final assessment combination, it is essential to understand the influence of the organic combination of each factor quantitatively reflected in the risk assessment. There are two primary methods for determining pipeline risk evaluation indicators: one is based on expert ratings and system safety theory [8–10], and the other is based on membership functions and fuzzy mathematics [11,12]. Methods of pipeline risk assessments can also be divided into two categories: evaluation through a simple qualitative method [13,14] and a semi-quantitative evaluation using a computer [15–20]. Although semi-quantitative evaluations can improve the subjective nature of qualitative evaluations, each factor's weight is primarily based on the analytic hierarchy process, which is influenced by human subjectivity. The weight of evidence (WOE)—genetic algorithm (GA)—backpropagation (BP) model is an information evaluation method supplemented by a BP neural network optimised by a GA, which can eliminate the influence of subjectivity on the weight of each factor, ensuring more objective evaluation results. In this study, the Sichuan-East Gas to Sichuan-Chongqing and Western Hubei pipeline route was considered as the research subject. Based on the survey data of on-site geological disasters, the WOE model was used to assess the danger along the pipeline, and then the BP neural network was employed to study the weight of each factor. Finally, based on the geographic information system (GIS), a geological disaster risk zoning map was obtained along the pipeline. The receiver

operating characteristic (ROC) curve was utilized to verify the reliability of the model and to provide reference information for other long distance pipelines.

## 2. WOE-GA-BP Model

### 2.1. Evidence Weight Method

The WOE law, a method based on Bayesian probability statistics [21], was first applied in medical diagnoses. During the 1990s, geologists Bonham-Carter et al. [22] and Ahterberg et al. [23] applied this method to mineral resource prediction. This method can avoid the subjective influence of weight factor assignment to a certain extent and considers the positive and negative weights of the index factors [24]. The calculation formula is as follows:

$$W_i^+ = \ln\left(\frac{P\{B/L\}}{P\{B/\overline{L}\}}\right) = \ln\left(\frac{P(B \cap L)/P(L)}{P(B \cap \overline{L})/P(\overline{L})}\right) = \ln\left[\frac{\frac{N_{pix1}}{N_{pix1}+N_{pix2}}}{\frac{N_{pix3}}{N_{pix3}+N_{pix4}}}\right] \tag{1}$$

$$W_i^- = \ln\left(\frac{P\{\overline{B}/L\}}{P\{\overline{B}/\overline{L}\}}\right) = \ln\left(\frac{P(\overline{B} \cap L)/P(L)}{P(\overline{B} \cap \overline{L})/P(\overline{L})}\right) = \ln\left[\frac{\frac{N_{pix2}}{N_{pix1}+N_{pix2}}}{\frac{N_{pix4}}{N_{pix3}+N_{pix4}}}\right] \tag{2}$$

$$W_{fi} = W_i^+ - W_i^-$$

$$\begin{cases} W_i^+ = \ln\left(\frac{P\{B/L\}}{P\{B/\overline{L}\}}\right) = \ln\left(\frac{P(B \cap L)/P(L)}{P(B \cap \overline{L})/P(\overline{L})}\right) = \ln\left[\frac{\frac{N_{pix1}}{N_{pix1}+N_{pix2}}}{\frac{N_{pix3}}{N_{pix3}+N_{pix4}}}\right] \\ W_i^- = \ln\left(\frac{P\{\overline{B}/L\}}{P\{\overline{B}/\overline{L}\}}\right) = \ln\left(\frac{P(\overline{B} \cap L)/P(L)}{P(\overline{B} \cap \overline{L})/P(\overline{L})}\right) = \ln\left[\frac{\frac{N_{pix2}}{N_{pix1}+N_{pix2}}}{\frac{N_{pix4}}{N_{pix3}+N_{pix4}}}\right] \\ W_{fi} = W_i^+ - W_i^- \end{cases} \tag{3}$$

where $P$ is the probability of a certain event, $P\{B/L\}$ is the probability of event $B$ under the condition that the $L$ event occurs, $B$ is the evaluation unit of the landslide in the secondary factor, $\overline{B}$ is the evaluation unit without landslides in the secondary factor, $L$ represents the evaluation unit with landslides in the study area, and $\overline{L}$ represents the evaluation unit with no landslides in the study area. $N_{pix1}$ represents the number of area grids in which landslides occurred within the second-level factor, $N_{pix2}$ represents the number of area grids in which landslides occurred outside the second-level factor, $N_{pix3}$ represents the number of grids in which landslides did not occur within the second-level factor, and $N_{pix4}$ represents the second-level factor. $N$ is the number of grids in which no landslides occurred outside of the factor.

$P\{B/L\}/P\{B/\overline{L}\}$ represents the sufficient rate of landslide occurrence, and $P\{\overline{B}/L\}/P\{\overline{B}/\overline{L}\}$ represents the necessity of landslide rate occurrence. $W_i^+$ represents the probability of a landslide under the influence of a secondary factor, which is a positive correlation weight, and $W_i^-$ represents the probability of a landslide that does not affect the secondary factor, which is a negative correlation weight. When $W_i^+ > 0$ or $W_i^- < 0$, the secondary factor is positively correlated with the occurrence of landslides. The larger the value, the stronger the correlation with landslides, and the higher the susceptibility to landslides. When $W_i^+ < 0$ or $W_i^- > 0$, the secondary factor is negatively correlated with the occurrence of landslides. The greater the negative value, the weaker the correlation with landslides, and the lower the susceptibility to landslides. $W_i^+ = 0$ or $W_i^- = 0$ indicates that this secondary factor is independent of the occurrence of a landslide. The difference between the two values of $W_{fi}$ represents the WOE. A larger value of $W_{fi}$ indicates that the promotion effect of the secondary impact factor on the occurrence of landslides is clearer. When $W_{fi} = 0$, the secondary factor is independent of the occurrence of landslides.

### 2.2. GA-BP Neural Network

The BP neural network has excellent multi-dimensional function mapping ability and is capable of reflecting the law of nonlinear problems [25], which can be used for pipeline risk assessment. In previous studies, BP neural networks have mostly been utilized for landslide risk assessments through the consideration of factors affecting landslide stability as the input layer, and those affecting risk index as the output layer. They were employed to predict the results based on the connection between neurons; however, the data were insufficient [26,27]. A GA was used to optimise the weights and thresholds of the neural network while using the network's nonlinear mapping capabilities, and improving its convergence speed and prediction accuracy [28]. Accordingly, the weight of each index factor in the BP neural network can be coupled with the weight of the evidence weight model to obtain the risk index of geological disasters along the pipeline. The GIS platform was employed to obtain the risk zone map and perform the calculation process, as shown in Figure 2.

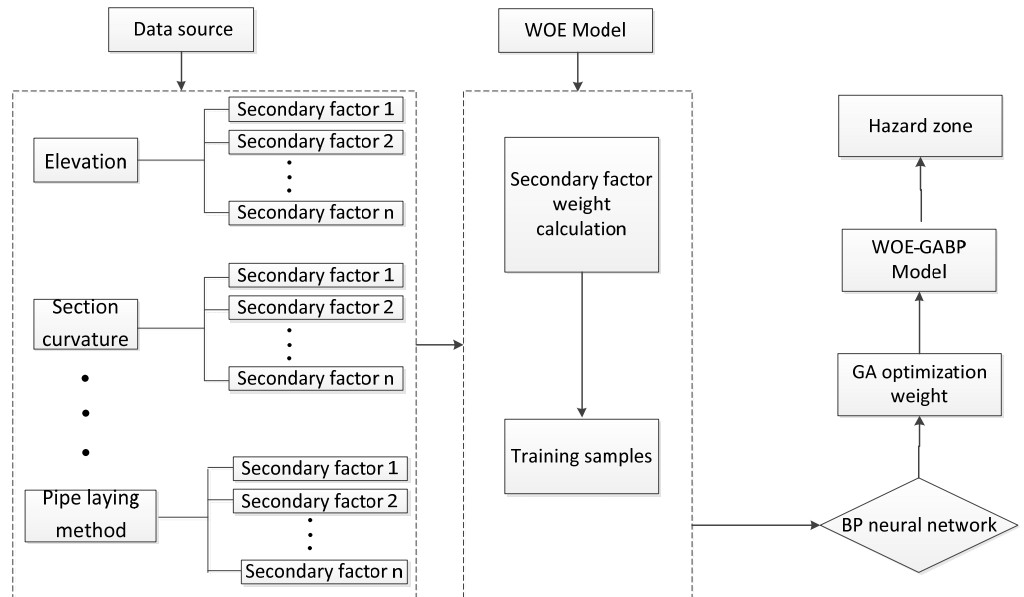

**Figure 2.** Technology roadmap.

## 3. Overview of the Study Area and Data Sources

### 3.1. Overview of the Study Area

The pipeline trunk line of the study area is 660 km long and passes through multiple stratigraphic sub-regions of Dazhou City, Chongqing City, Enshi Prefecture, and Yichang City. Controlled by geological structures, the strata along the pipeline have complex lithologic characteristics and diverse rock mass types. Mudstone, limestone, clastic rock, and carbonate rock are the main rock types in this region. From west to east, it passes through the third subsidence zone of the New Cathaysian System and central zone of the Yangtze Quasi-platform. It belongs to a subtropical humid monsoon climate zone, characterized by cold winters, hot summers, heavy rains in spring and autumn, high humidity, and abundant rainfall. The average temperature is 15–17 °C. The landforms along the route are river valleys, hills, and mid-mountain landforms; accompanied by numerous geological disasters such as collapses, landslides, and unstable slopes (Figure 3).

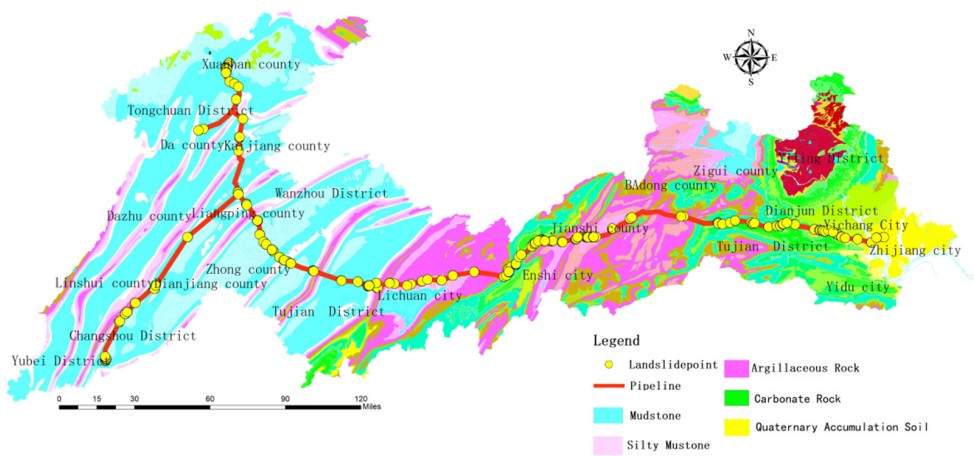

**Figure 3.** Hazard point and lithology map.

*3.2. Data Source*

Based on the geological disaster survey data along the pipeline and remote sensing data, a regional landslide disaster database was established to analyse and map the landslide disaster along the pipeline. The main data included: (1) DEM data along the pipeline to analyse basic information, such as elevation, slope, aspect, and curvature of the study area; (2) a 1:50,000 geological map along the pipeline to advance the formation information of the study area; (3) satellite remote sensing data along the pipeline to extract vegetation coverage in the study area; (4) landslide survey and survey data along the pipeline along with 91 satellite maps to determine the area and distribution of landslides.

*3.3. Characteristics of Landslides in the Study Area*

The survey results indicated 220 disaster points along the pipeline, including 135 unstable slopes, 44 collapses, and 41 landslides. Landslides and unstable slopes are the main geological disasters along the pipeline, accounting for more than 80% of the geological disasters, with a total area of 380,000 km². The largest and smallest landslide areas are 4.5 km² and 19 m² (Figure 4).

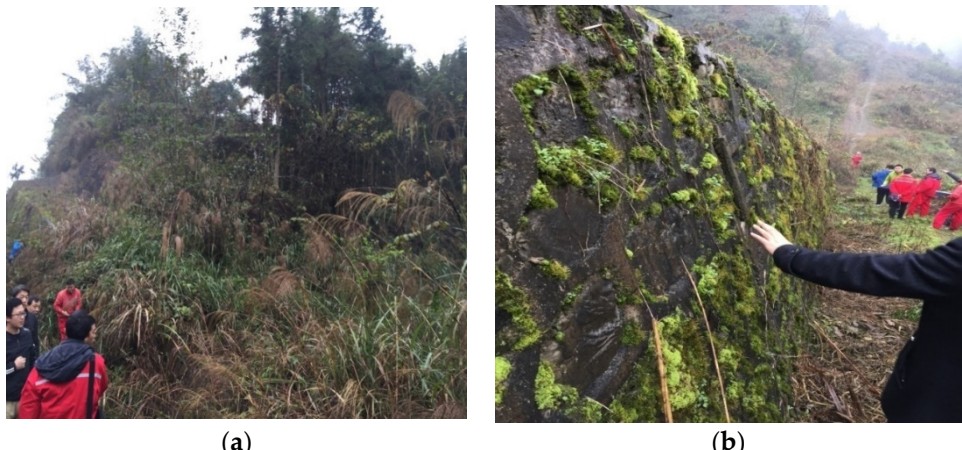

(**a**)                    (**b**)

**Figure 4.** (**a**) XP-17-36 overall view and (**b**) cracking of retaining wall.

## 4. Secondary Index Factor Classification and Weight Calculation

Pipeline hazard research differs from general geological hazard susceptibility research. While considering factors affecting landslide development, the pipeline laying method should be considered due to its significant relationship with pipeline force [29]. Based on the on-site survey, we considered eight factors as evaluation indicators on the basis of correlation tests: elevation, slope, aspect, ground curvature, lithology, rainfall, normalised

difference vegetation index (NDVI), and laying methods. Second-order factors were divided based on WOE, and the resolution of grid cells was 5 m × 5 m.

### 4.1. Index Factor Classification

Index factors are divided into discrete and continuous types based on the degree of influence on the landslide. For continuous data, the factors are first discretised; for discrete data, each grade has a clear physical significance. The degree of influence of the secondary state of each indicating factor on landslides was evaluated by the landslide area ratio, graded area ratio, and evidence weight [30]. The landslide area ratio is the landslide area occurring in the secondary state of the index factor compared to the total landslide area in the entire region, and the graded area ratio is the area of each secondary state of the index factor compared to the total area of the index factor. The relative size of the two ratios characterises the importance of the secondary state classification of the index factor for landslide susceptibility [31]. If the landslide area ratio is greater than the graded area ratio, the landslide is prone to occur within the classification of the state, which is a rare occurrence.

#### 4.1.1. Elevation

The development of landslides is closely related to their distribution elevation. On the one hand, the terrain gradient is different at different elevations, which results in a difference in the collecting capacity of surface water. On the other hand, the intensity of human engineering activities is different in different elevation ranges, which results in a difference in surface conditions; hence, elevation is an important factor in the landslide disaster-inducing environment. The elevation range of the study area is 17–1937 m, of which more than 95% of landslides are distributed in the range of 17–1500 m, with no landslides beyond 1500 m. According to the actual distribution, the elevations are 37–500 m, 500–1000 m, 1000–1500 m, and 1500–1974 m. A total of five secondary states constitute the number of grids in each region, number of geological hazard points in each region, and number of grids occupied by geological hazard points (Figure 5).

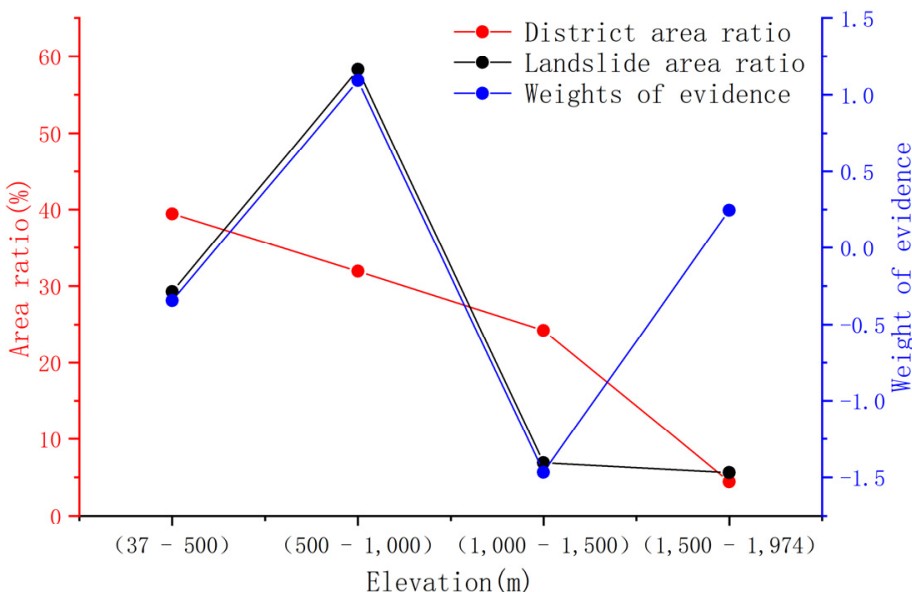

**Figure 5.** Elevation statistical analysis results.

#### 4.1.2. Slope

The slope affects surface water collection, groundwater infiltration, groundwater flow direction, and stress distribution, and consequently, the gestation and development of landslides. Therefore, slope is an important factor that affects the occurrence, development, and morphological characteristics of landslides. The slope of the terrain in the study area

ranged from 0° to 83°, and landslide slope was distributed between 0° and 30°, accounting for nearly 90% of the area. The slope was discretised with a step length of 10°, and the landslide area ratio in each secondary state, classification area ration, and WOE were calculated (Figure 6).

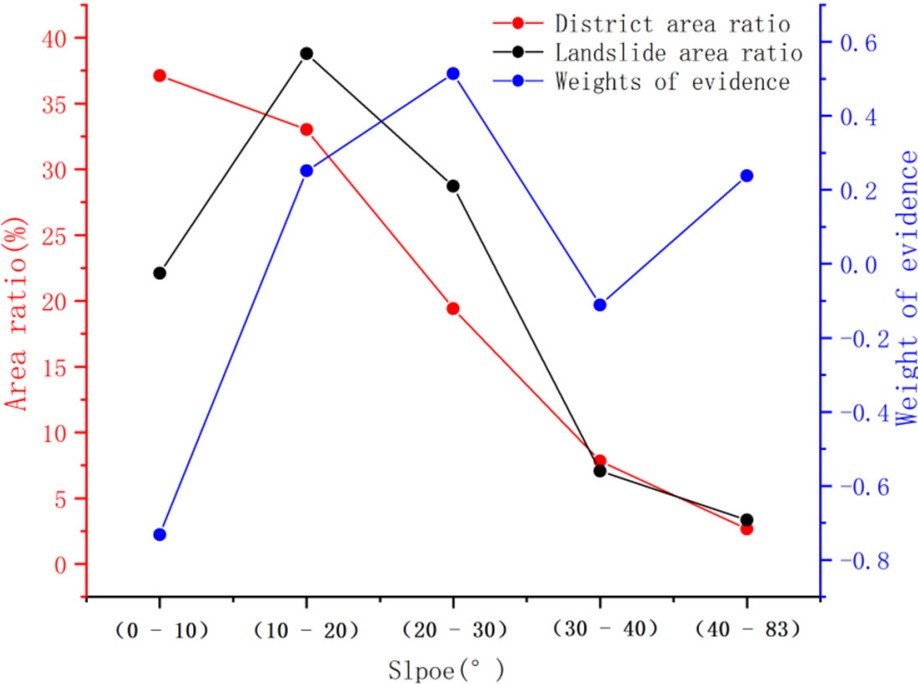

**Figure 6.** Slope statistical analysis results.

### 4.1.3. Aspect

Owing to variations in physical conditions, such as sunlight and climate on different slopes, the resistance to weathering and water evaporation on various slopes differ. Generally, sunny slopes and slopes affected by monsoons throughout the year are more prone to landslides and other geological disasters. Based on the principle of 45° area and easy software calculation, eight areas were utilized in this research: 0°–45°, 45°–90°, 90°–135°, 135°–180°, 180°–225 °, 225°–270°, 270°–315°, and 315°–360° (Figure 7).

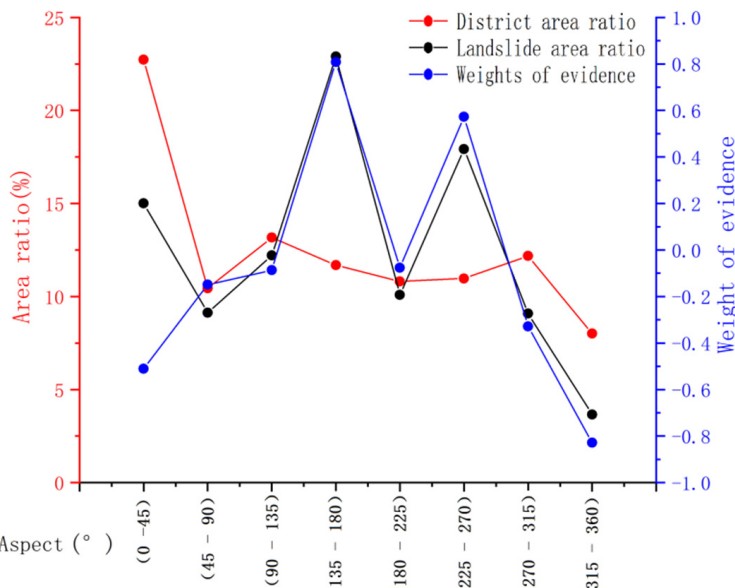

**Figure 7.** Aspect statistical analysis results.

### 4.1.4. Section Curvature

Section curvature is the rate of change of the slope or aspect in a certain direction. The magnitude of curvature reflects the undulating shape of the ground from the side and magnitude of the slope of the ground. Under the influence of long-term physical and chemical effects, the rock is broken down and a large amount of fragmented rock and gravel soil accumulates on the slope's surface. Similarly, due to internal action in the earth (such as volcanic eruptions and earthquakes), slopes with a particular potential energy lose their stability, causing landslides and other geological disasters. The areas of landslide development in the distribution areas of concave and convex slopes were similar; however, the susceptibility to landslides differed.

### 4.1.5. Stratum Lithology

The stratum plays an important role in landslide formation and development. However, different types of engineering geological rock groups have different degrees of influence on the formation of landslides and other geological disasters. Moreover, engineering geological rock sets determine the type and scale of landslide characteristics. Based on the national 1:50,000 geological map and a previous investigation, the lithology of the study area was divided into five types: mudstone, silty mudstone, argillaceous rock, carbonate rock, and quaternary accumulation soil, and the statistics of the lithology landslide area ratios were evaluated, among other quantities (Figure 8).

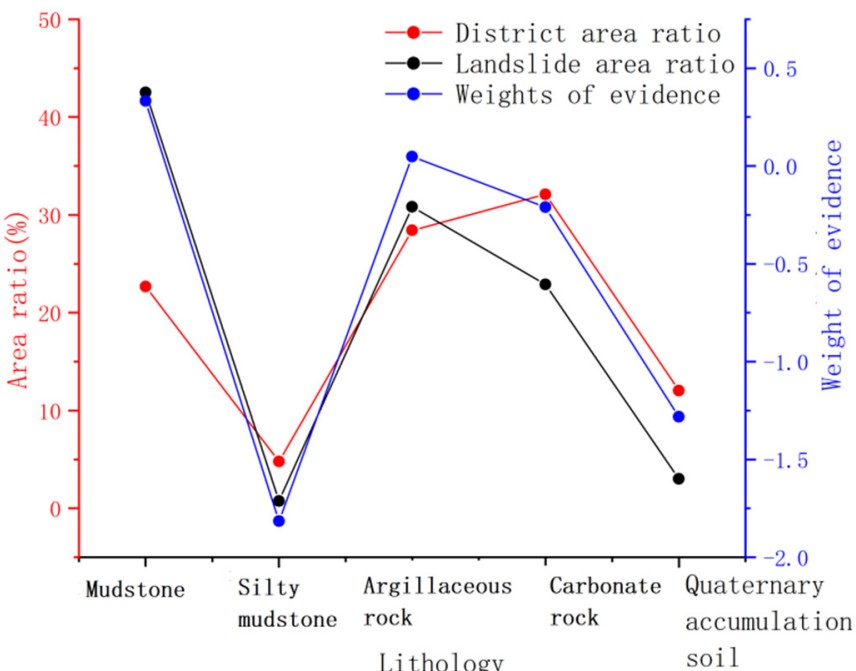

**Figure 8.** Lithology statistical analysis results.

### 4.1.6. Rainfall

The effects of rainfall on landslides are reflected in several aspects. First, rainwater penetrates into fracture surfaces or cracks due to gravity, which reduces the frictional resistance between the contact surfaces. In addition, water content significantly affects the mechanical properties of the soil. When the sliding force is greater than the anti-sliding force, the slope becomes unstable. Based on the national average rainfall distribution map in 2018, the rainfall amounts in the study area were divided into three categories: 1000–1200 mm, 1200–1600 mm, and >2000 mm (Figure 9).

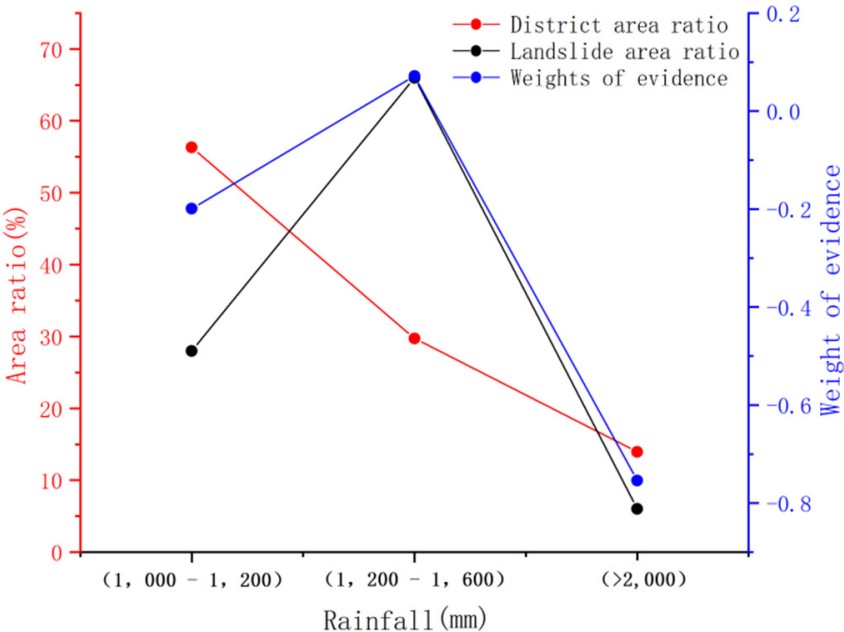

**Figure 9.** Rainfall statistical analysis results.

### 4.1.7. NDVI

The vegetation coverage rate significantly affects the stability of the slope. On the one hand, vegetation coverage reduces the ground runoff and rainwater infiltration along with the effects of rainfall on slope stability. On the other hand, vegetation roots can reinforce slopes and increase slope stability. Based on Landsat8 images, the NDVI index in the study area was extracted in this study to elucidate its effect (Figure 10).

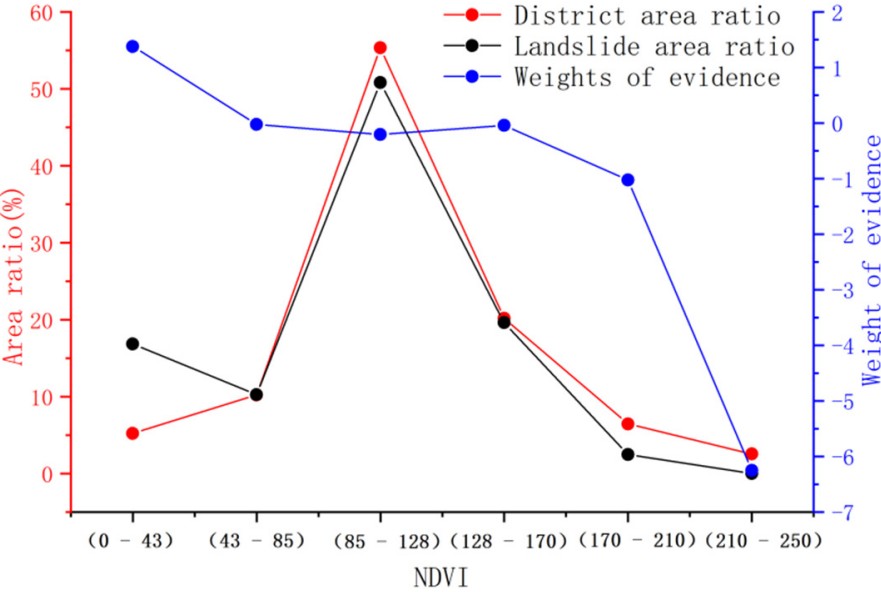

**Figure 10.** NDVI statistical analysis results.

### 4.1.8. Pipe-Laying Method

Unlike in general landslide body susceptibility studies, the pipeline-laying method is employed to study pipelines. This method significantly affects the pipeline stress conditions, and the pipeline stress deformation degree directly affects the pipeline operational safety; thus, pipeline laying is the key factor in evaluating the risk of pipelines. According to the relationship between pipeline direction and the main sliding direction of the landslide, the laying methods can be divided into three types: horizontal, longitudinal, and oblique.

According to Zhang et al. [29], horizontal laying is used when the pipe laying stress is the largest, which is a critical situation; hence, in the evaluation analysis, horizontal laying was the highest, and vertical laying was the lowest. It can be seen from Figure 11 that the higher the value of laying method, the greater the contribution to the risk of landslides, which is consistent with the research results of Zhang et al. [29].

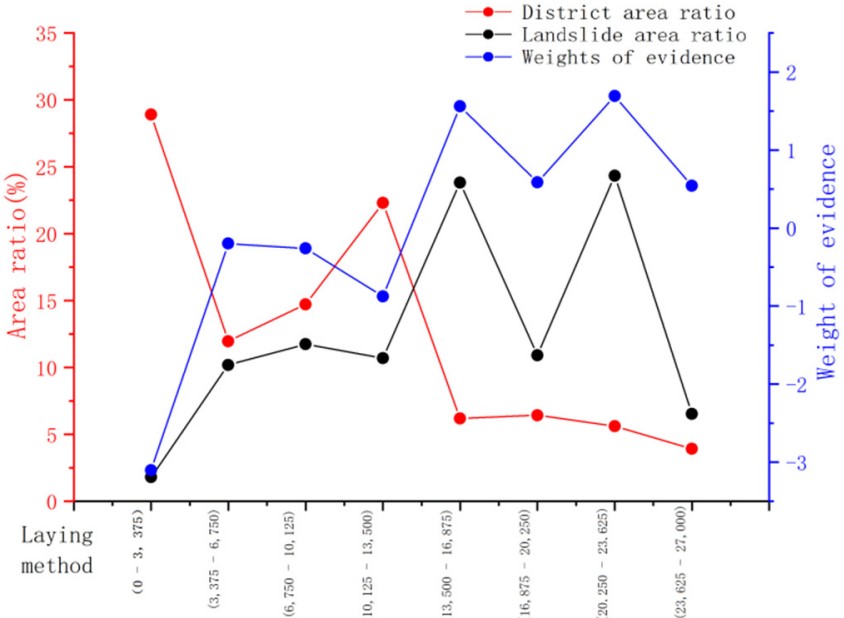

**Figure 11.** Statistical analysis results of laying method.

### 4.2. GA-BP Model Construction Analysis Based on WOE Model

### 4.2.1. WOE Model

Equation (1) was used to calculate the WOE for each secondary factor (Table 1). Risk zoning along the pipeline was performed using the ArcGis grid stacking tool (Figure 12). A total of 273,320,894 grids were present along the pipeline, including 83,023,949 high-risk landslide grids, 133,985,984 medium-risk grids, and 56,310,961 low-risk grids, accounting for 30.38%, 49.02%, and 20.6% of all grids, respectively (Table 2).

### 4.2.2. GA-BP Model Construction and Weight Calculation

The neural network model represents a black box with nonlinear characteristics. A single-layer neural network model can uniformly approximate any continuity function [32]. Therefore, the model was constructed using a single hidden layer based on the index factor. The number of neurons in the input layer $N_i$ was eight, and the number of neurons in the output layer $N_o$, also known as risk index, was one. However, the optimal number of hidden layers is affected by many other factors and is often difficult to determine. According to Hecht-Nielsen [33] and Lawrence and Fredrickson [34], the recommended upper limit of the hidden node is $2N_i + 1$ and the lower limit is $(N_i + N_o)/2$. Therefore, the number of hidden layer neurons considered in this study should be in the range of 5–17. After comparing the size of the neural network mean square error, the number of hidden layer neurons was determined to be 15, based on which a neural network of 8–15–1 was constructed.

The sample size used to train the neural network should be neither too large nor too small. Widrow [35] conducted a special study on the number of training samples and proposed the 'rule of thumb'. Baum and Haussler [36] provided a more specific range of the number of training samples $\alpha$; that is, $W/\gamma < \alpha < (W/\gamma)\log(N/\gamma)$, where N is the number of neurons and W is the number of ownership values. Therefore, the appropriate sample size for this study was in the range of 1350–3213.

**Table 1.** Index factors and $W_{fi}$.

| Index Factors | Index Classification | $W_{fi}$ | Index Factors | Index Classification | $W_{fi}$ |
|---|---|---|---|---|---|
| Elevation (m) | 37–500 | −0.345 | Stratum lithology | Mudstone | 0.332 |
| | 500–1000 | 1.092 | | Silty mudstone | −1.815 |
| | 1000–1500 | −1.46 | | Argillaceous rock | 0.047 |
| | 1500–1974 | 0.246 | | Carbonate rock | −0.211 |
| Slope (°) | 0–10 | −0.732 | | Quaternary accumulation soil | −1.282 |
| | 10–20 | 0.251 | NDVI | 0–43 | 1.379 |
| | 20–30 | 0.514 | | 43–85 | −0.023 |
| | 30–40 | −0.111 | | 85–128 | −0.204 |
| | 40–83 | 0.238 | | 128–170 | −0.038 |
| Aspect (°) | 0–45 | −0.511 | | 170–210 | −1.023 |
| | 45–90 | −0.149 | | 210–250 | −6.249 |
| | 90–135 | −0.087 | Pipe laying method | 0–3375 | −3.103 |
| | 135–180 | 0.808 | | 3375–6750 | −0.199 |
| | 180–225 | −0.077 | | 6750–10,125 | −0.262 |
| | 225–270 | 0.573 | | 10,125–13,500 | −0.875 |
| | 270–315 | −0.328 | | 13,500–16,875 | 1.561 |
| | 315–360 | −0.828 | | 16,875–20,250 | 0.588 |
| Section curvature | >0 | −0.898 | | 20,250–23,625 | 1.692 |
| | <0 | 0.898 | | 23,625–27,000 | 0.541 |
| Rainfall (m) | 1000–1200 | −0.199 | | | |
| | 1200–1600 | 0.0716 | | | |
| | >2000 | −0.754 | | | |

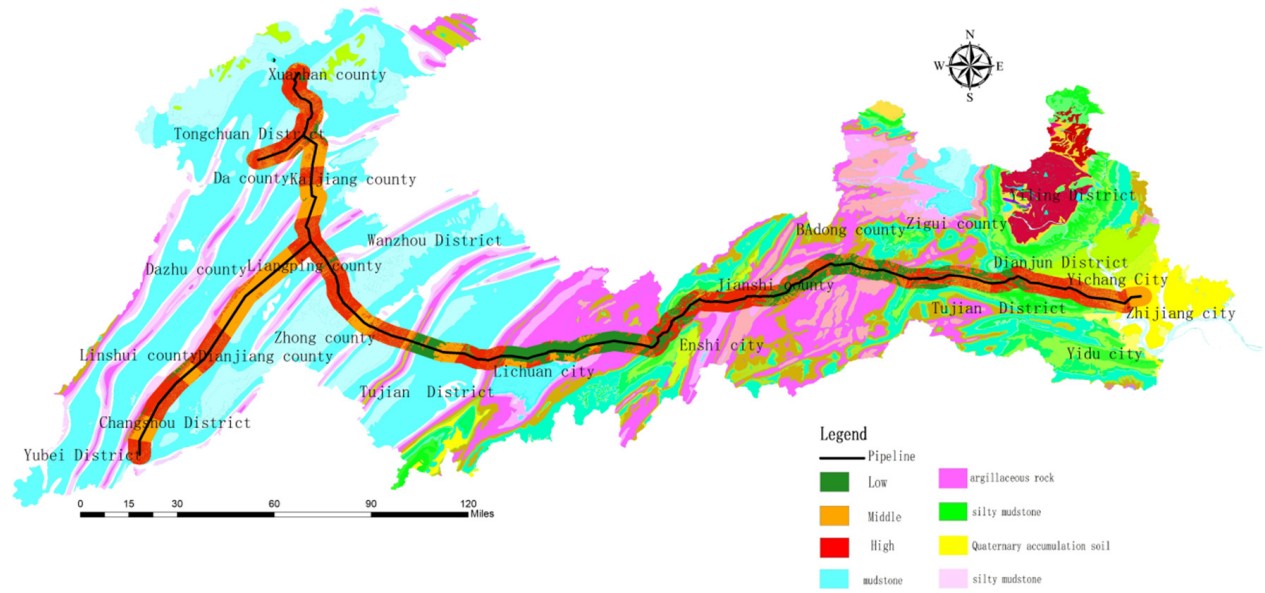

**Figure 12.** Hazard zoning map.

**Table 2.** Hazard zoning statistics table.

| Hazard Zoning | Number of Grids | Proportion of Each District | Actual Area (km²) | Main Lithology and Geomorphic Characteristics |
|---|---|---|---|---|
| High | 83,023,949 | 30.38% | 2076 | Mudstone, deep valleys, and steep terrain |
| Middle | 133,985,984 | 49.02% | 3350 | Siltstone, silty mudstone, low mountain, and hill landform |
| Low | 56,310,961 | 20.6% | 1408 | Limestone, sandstone, and low mountain area appearance |

In this study, we used 273,320,894 grid units and 392,572 landslide grid units in the area along the pipeline. A total of 1500 random landslide grids and non-landslide grids were selected, and a total of 3000 grids were used as training samples. Index factor attributed values in the grid were extracted as input-layer data, and risk index as output-layer data. Firstly, the BP neural network training sample (Figure 13) was used to calculate the weight value, which was then optimised using the GA to improve the BP neural network. Finally, the network was retrained (picture b) to calculate the optimised weight value (Table 3).

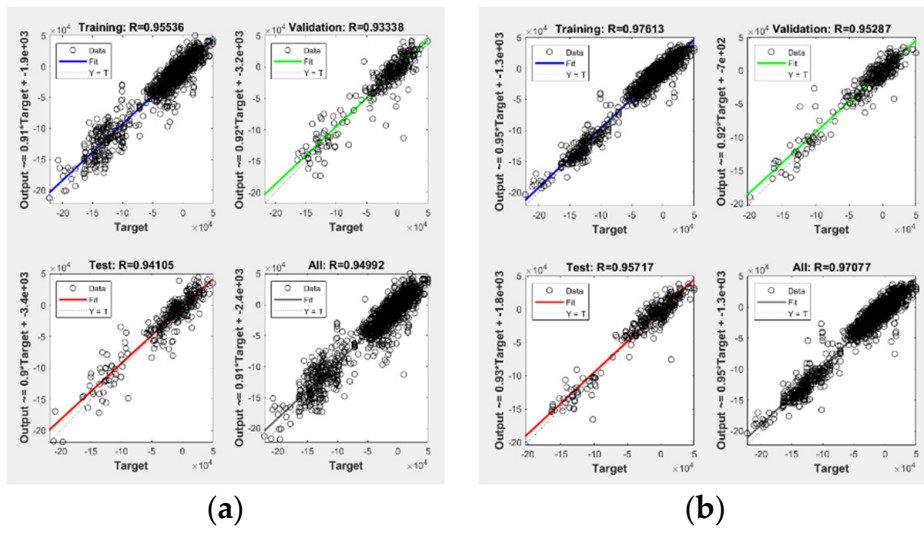

(**a**)　　　　　　　　　　　　　　　(**b**)

**Figure 13.** Correlation coefficients of neural network training: (**a**) BP and (**b**) GA-BP.

**Table 3.** Calculated index factor weights.

| Serial Number | Index Factor | BP Weight | GA-BP Weight |
|---|---|---|---|
| 1 | Elevation | 0.97709 | 0.721317 |
| 2 | Slope | 0.979567 | 0.805494 |
| 3 | Aspect | 1.017634 | 0.788956 |
| 4 | Section curvature | 0.986155 | 0.820928 |
| 5 | Stratum lithology | 1.031557 | 1.049514 |
| 6 | Rainfall | 0.999602 | 0.75175 |
| 7 | NDVI | 1.000239 | 0.962203 |
| 8 | Pipe laying method | 1.047163 | 1.02511 |

Figure 13 shows that, in both the BP and GA-BP neural network cases, the R correlation coefficient is greater than 0.95 after training, indicating that the input and output layers are highly correlated. In addition, the value of R for the BP neural network optimised by GA increases by nearly 0.02, indicating that the optimisation improved the BP model.

## 5. Slope Risk Evaluation and Accuracy Analysis

### 5.1. Geological Hazard Assessment

Based on the ArcGIS raster analysis tool and the WOE model evaluation, each indicator factor can be multiplied times the weights obtained by training the BP and GA-BP models in Table 3, and re-analysed to obtain the risk zone map.

Comparing the hazard zone results obtained by the WOE model (Figure 14), WOE-BP model (Figure 15), and WOE-GA-BP model (Figure 16), the following conclusions can be drawn:

(1) High and medium-risk areas are primarily distributed in the regions of argillaceous rocks, deep valleys, and steep terrain, because the mechanical properties of argillaceous rocks are weaker than those of other rocks and the steep terrain can promote the occurrence of a slope.

(2) According to the weight calculation results, the stratum lithology, pipe laying method, and NDVI are the main factors affecting the hazard zone. The calculation results for the stratum lithology and NDVI weights are consistent with the knowledge of geological experts on the weights of important controlling factors, and the calculation results of the weights of the laying method are consistent with the analysis of the pipeline forces performed by Zhang et al. [29].

(3) Compared to hazard zone maps obtained using the other two models, those acquired by applying the WOE-BP model have a larger proportion of high-prone areas and contain most of the historical landslide points, hence having the most practical significance.

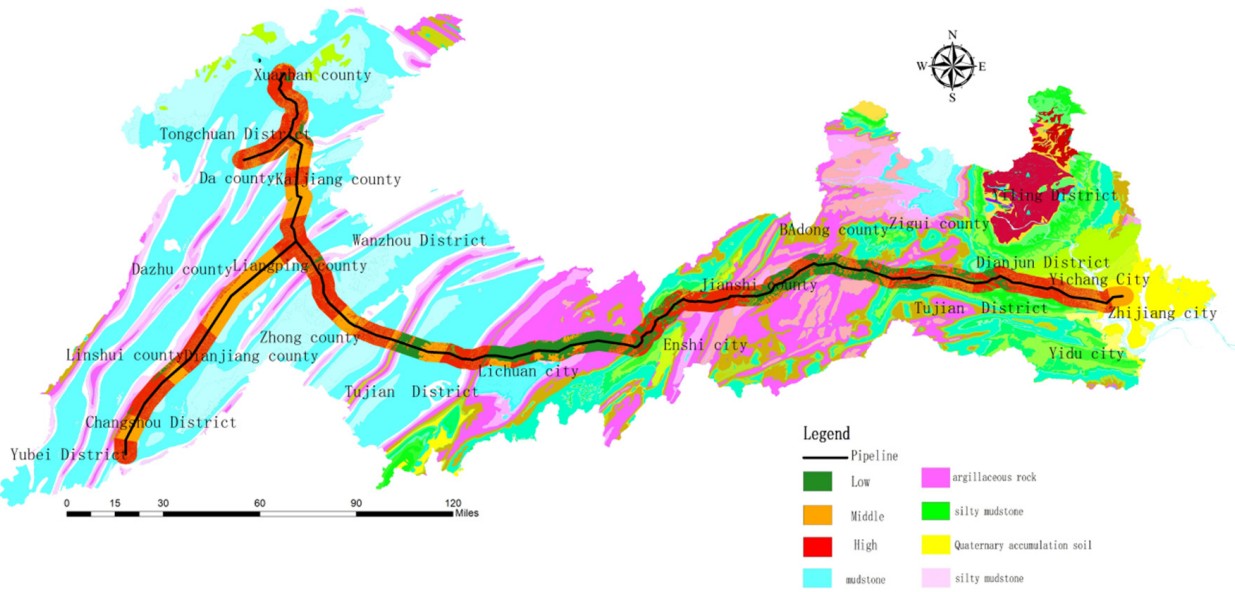

**Figure 14.** Hazard partition of the WOE model.

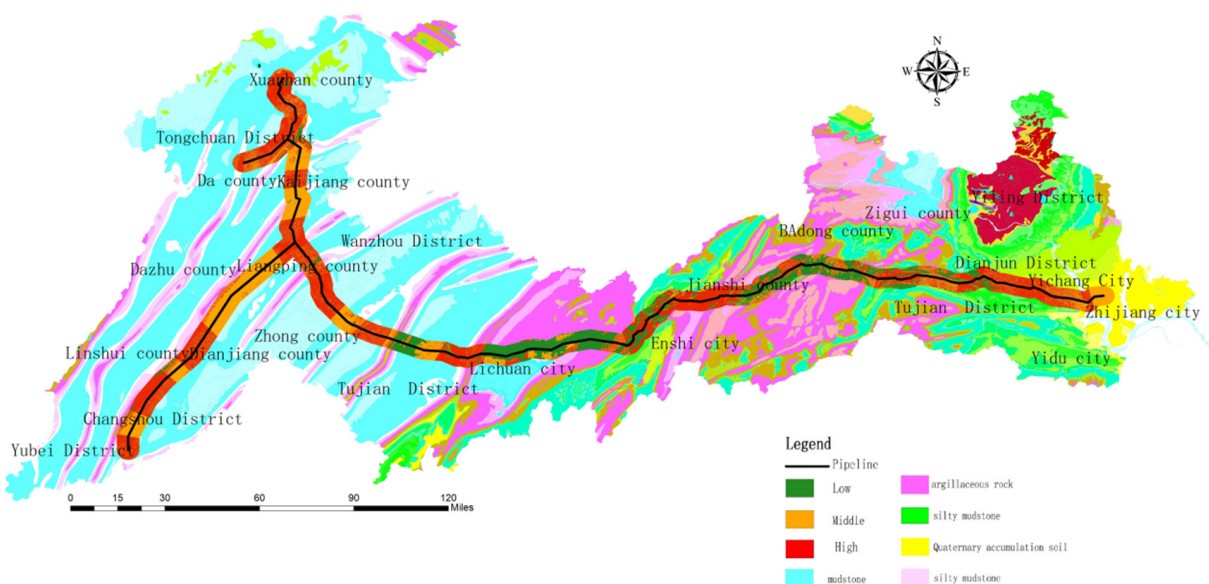

**Figure 15.** Hazard partition of the WOE-BP model.

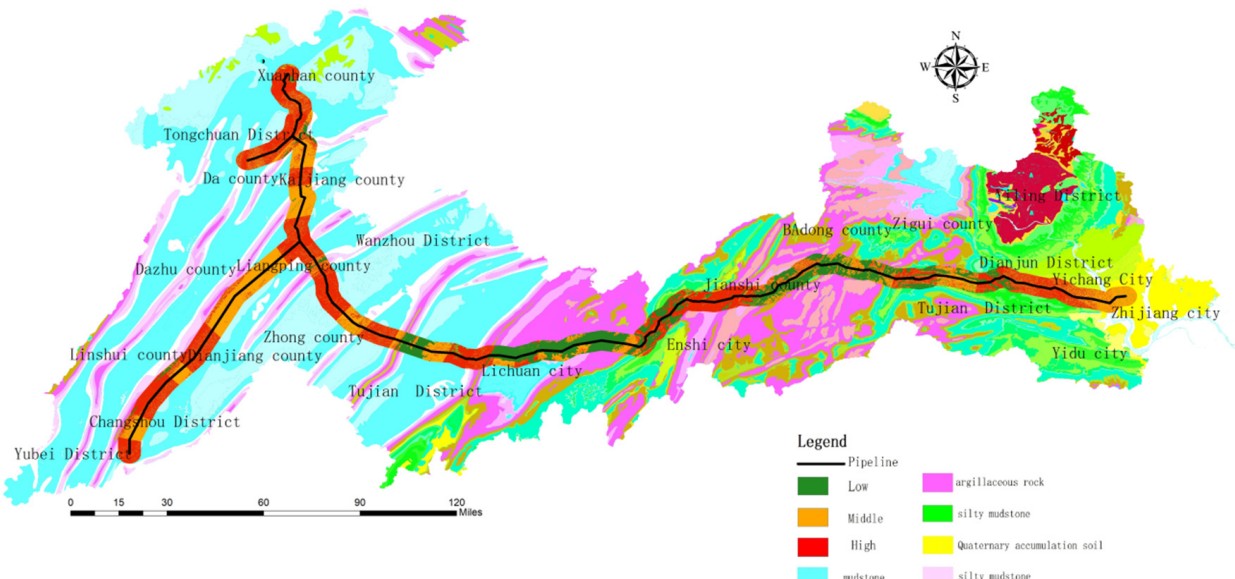

**Figure 16.** Hazard partition of the WOE-GA-BP model.

*5.2. Accuracy Evaluation*

The area under the ROC curve (AUC) was employed as the measurement standard. Fawcett [37] discussed the basic theories and methods of calculating the ROC curve and AUC.

When the AUC is less than 0.7, the model evaluation accuracy is poor; meanwhile, when the AUC is between 0.7 and 0.8, the model evaluation accuracy is medium, and when the AUC is greater than 0.8, the model evaluation accuracy is good. In this study, the hazard indices obtained from the three models were divided into 100 intervals from large to small, and the occurrence frequency of historical landslides gradually decreased. ROC curves of the three models were drawn with the total grid frequency in the study area as the abscissa, and historical occurrence frequency within the grid as the ordinate (Figure 17).

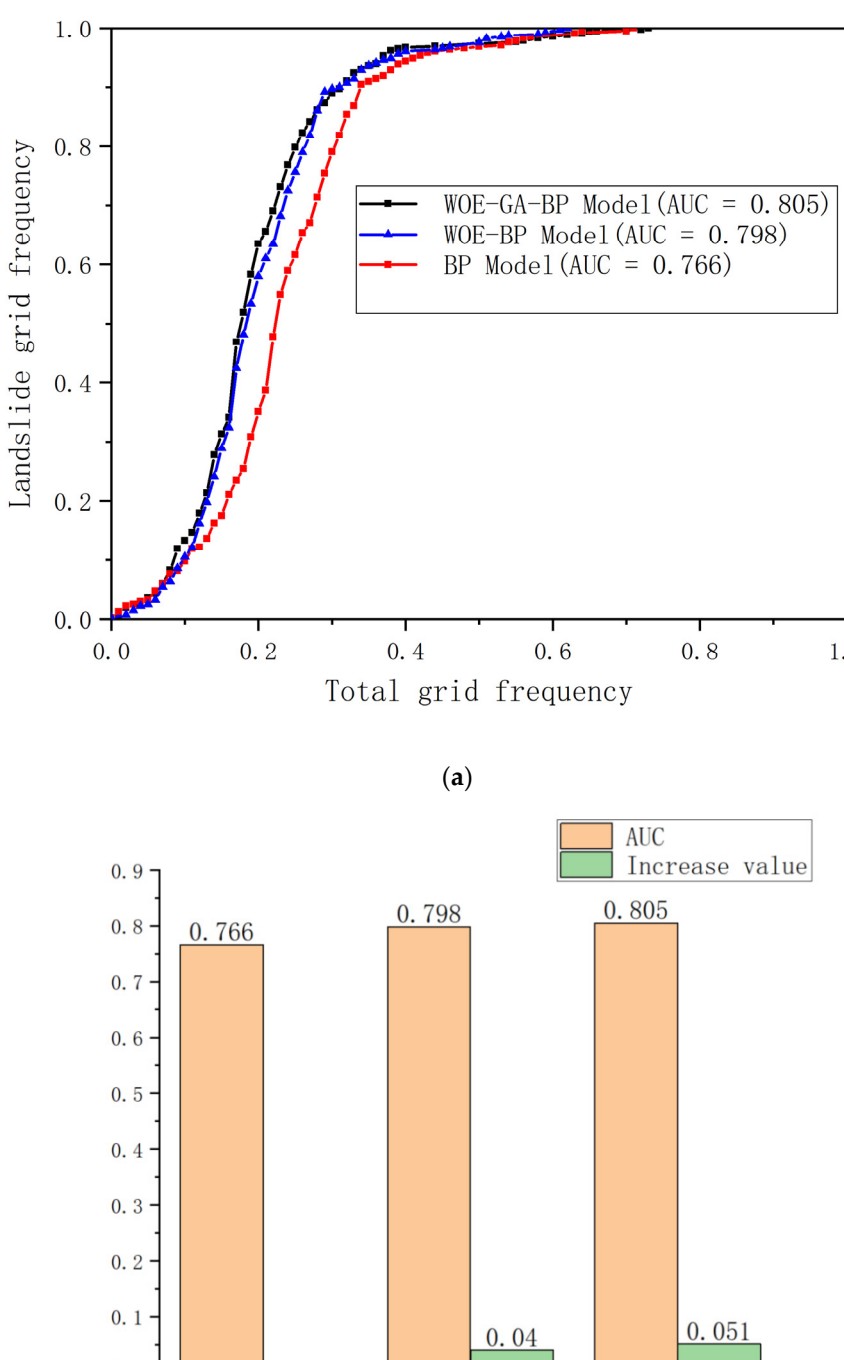

**Figure 17.** (**a**) ROC curve. (**b**) accuracy comparison chart.

According to the ROC curve, the AUCs of the three models are 76.6%(WOE Model), 79.8%(WOE-BP Model), and 80.5%(WOE-GA-BP Mod); however, only the area corresponding to the WOE-GA-BP model exceeds 0.8, resulting in good evaluation accuracy, and indicating that the predictive evaluation ability of the WOE-GA-BP model is better than those of the WOE and WOE-BP models.

## 6. Conclusions

In this study, we developed a WOE-GA-BP model to predict the occurrence of geological disasters and prevent the damage to pipelines using the GIS at the risk site. The work conducted in this investigation can be summarised as follows:

(1) The area along a certain gas pipeline was considered as the research object. Eight evaluation factors, specifically, the elevation, slope, aspect, ground curvature, lithology, rainfall, NDVI, and laying method, were selected to establish a geological hazard risk assessment system along the pipeline. The risk assessment of the study area was conducted based on GIS and WOE-GA-BP models. The results were consistent with the actual situation. The AUC reached 80.5%, indicating that the WOE-GA-BP model is an effective risk assessment and prediction model.

(2) In contrast to the risk assessment of regional landslides, we assessed laying methods closely related to the force of the pipeline. According to the analysis of the WOE model, the WOE for horizontal laying was 1.692, which was the largest. Meanwhile, in the GA-BP model analysis, the weight of the laying method was approximately 1.04, indicating a positive correlation between the stress on the pipeline and risk. This also provides suggestions for pipeline laying. In addition, we observed that during the construction of a new line, it is necessary to pass through landslides and unstable slope areas with vertical or diagonal paving as far as possible to reduce risk.

(3) The GA-BP model was used to calculate the weight of each index factor. The results showed that the laying method, stratum lithology, and NDVI were the influencing factors for risk assessment.

This is consistent with the actual distribution of landslides and unstable slopes, indicating that the GA-BP model can effectively calculate the weights of index factors, avoid subjective effects of previous calculation methods, and utilize the advantages of data mining and machine learning.

**Author Contributions:** Conceptualization, B.H. and M.B.; methodology, B.H.; software, B.H.; validation, H.S. and M.B.; formal analysis, B.H.; investigation, Y.L. and X.L.; resources, B.H and X.L.; data curation, B.H.; writing—original draft preparation, B.H.; writing—review and editing, B.H.; visualization, Y.Q.; supervision, M.B.; project administration, M.B.; funding acquisition, M.B. All authors have read and agreed to the published version of the manuscript.

**Funding:** This research was funded by the National Key Research and Development Project of China, the Ministry of Science and Technology of China (grant 2018YFC1505500).

**Institutional Review Board Statement:** Not applicable.

**Informed Consent Statement:** Informed consent was obtained from all subjects involved in the study.

**Data Availability Statement:** Data are available upon reasonable request to the corresponding authors.

**Conflicts of Interest:** The authors declare no conflict of interest.

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
