# Peer review of "Risk Assessment of Pipeline Engineering Geological Disaster Based on GIS and WOE-GA-BP Models"

_applsci, doi:10.3390/app11219919_

Round 1

Reviewer 1 Report

The research topic is good and interested for reader.

The research is suitable for the journal.

The written English seems like translated directly from Chinese writing. Suggest pay more attention on the order of words in a text.

Line 112-116, The pix1-pix4 in the Npix should be subscript.

Line 119-127, The i in the Wi should be subscript.

Author Response

1. The English language and style have been modified by the English editor.

2. The font and image resolution in Figures 3, 12, 14, 15, 16, and 17 have been adjusted to better present the research results.

3. The corresponding words in line 112-116 and  119-127 have been changed to subscripts.

Reviewer 2 Report

The authors followed a good mix of probabilistic risk assessment, genetic algorithm, and neural network.

I recommend accept to publish with some revision as stated above. 

Author Response

1. The English language and style have been modified by the English editor.

2. In the introduction, the research background of the indicator system has been added, and a total of 5 references from 8 to 12 have been added.

3. The font and image resolution in Figures 3, 12, 14, 15, 16, and 17 have been adjusted to better present the research results.
